# HisCoM-G×E: Hierarchical Structural Component Analysis of Gene-Based Gene–Environment Interactions

**DOI:** 10.3390/ijms21186724

**Published:** 2020-09-14

**Authors:** Sungkyoung Choi, Sungyoung Lee, Iksoo Huh, Heungsun Hwang, Taesung Park

**Affiliations:** 1Department of Applied Mathematics, Hanyang University (ERICA), Ansan 15588, Korea; day0413@hanyang.ac.kr; 2Center for Precision Medicine, Seoul National University Hospital, Seoul 03080, Korea; biznok@snu.ac.kr; 3Department of nursing, College of Nursing and Research Institute of Nursing Science, Seoul National University, Seoul 03080, Korea; huhixoo@gmail.com; 4Department of Psychology, McGill University, Montreal, QC H3A 1G1, Canada; heungsun.hwang@mcgill.ca; 5Department of Statistics, Seoul National University, Seoul 08826, Korea; 6Interdisciplinary Program in Bioinformatics, Seoul National University, Seoul 08826, Korea

**Keywords:** gene–environment interactions, generalized structured component analysis (GSCA), genome-wide association study (GWAS)

## Abstract

Gene–environment interaction (G×E) studies are one of the most important solutions for understanding the “missing heritability” problem in genome-wide association studies (GWAS). Although many statistical methods have been proposed for detecting and identifying G×E, most employ single nucleotide polymorphism (SNP)-level analysis. In this study, we propose a new statistical method, Hierarchical structural CoMponent analysis of gene-based Gene–Environment interactions (HisCoM-G×E). HisCoM-G×E is based on the hierarchical structural relationship among all SNPs within a gene, and can accommodate all possible SNP-level effects into a single latent variable, by imposing a ridge penalty, and thus more efficiently takes into account the latent interaction term of G×E. The performance of the proposed method was evaluated in simulation studies, and we applied the proposed method to investigate gene–alcohol intake interactions affecting systolic blood pressure (SBP), using samples from the Korea Associated REsource (KARE) consortium data.

## 1. Introduction

Common genetic variants, identified by genome-wide association studies (GWAS), generally explain only a very small proportion of the variance of complex diseases or traits [1,2]. This may be because the genetic risk of single nucleotide polymorphisms (SNPs) is modified by environment, gender, age, or other factors. Understanding those interaction effects can help us to explain the “missing heritability” problem of complex diseases or traits [2]. Several studies of gene–environment interactions (G×E) have been performed for a variety of human complex diseases [3,4,5]. However, these studies have been mainly for candidate disease-associated genes. Moreover, findings from G×E studies based on genome-wide scale investigations [6] are quite limited, due to statistical power issues. An implementation of HisCoM-G×E can be downloaded from the website http://statgen.snu.ac.kr/software/hiscom-gxe.

Several methods have been proposed to improve statistical power for identifying G×E. These methods can be broadly categorized into two types. The first, which aims at increasing statistical power by performing an SNP-based G×E test, is the two-stage approach [7,8]. In the first stage, filtering is conducted at the SNP level. In the next stage, SNP-based G×E tests are conducted, and correctional procedures for multiple testing are performed only on the remaining SNPs.

The second type of method focuses on increasing statistical power by performing gene-based (or set-based) G×E analysis. The gene-based G×E test has several advantages, including aggregating multiple G×E signals within the same gene and reducing multiple-testing burdens. Furthermore, the gene-based G×E test simplifies biologically meaningful interpretation of the disease. Ma et al. proposed that gene-based gene–gene interaction (GGG) tests represent an extension of four existing methods of combing *p*-values [9] as follows: (1) gene-based association test using extended Simes (GATES) procedures [10]; (2) truncated tail strength [11]; (3) truncated *p*-value product [12]; and (4) minimum *p*-value [13]. GGG tests have been introduced for gene–gene interaction assessments at the gene level, and can also be applied to G×E studies.

Additionally, Lin et al. (2013) proposed a gene-based G×E test, i.e., the gene–environment set association test (GESAT), by extending the SNP-set (sequence) kernel association test (SKAT) to a G×E setting [14]. Furthermore, Lin et al. (2016) proposed the interaction sequence kernel association test (iSKAT), for assessing rare variants by environmental interactions [15]. The GESAT and iSKAT methods assume random G×E effects, following a distribution with mean zero and variant *τ*^2^; therefore, testing for G×E effects is equivalent to testing for a zero variance of *τ*^2^. The iSKAT method can also be applied to G×E studies, for common variants, using the weight parameter, *w* = Beta (minor allele frequency (MAF), 0.5, 0.5) [15,16]. 

In this study, we present a novel statistical method for G×E analysis, namely the Hierarchical structural CoMponent analysis of gene-based Gene–Environment interactions (HisCoM-G×E). The proposed method is based on generalized structured component analysis (GSCA), which tests hypothesized defined latent variables as components, using the weighted sums of observed variables [17]. Taking that into account, Lee et al. (2016) proposed a pathway-based approach, using a hierarchical structure of collapsed rare variants of high-throughput sequencing data (PHARAOH), by extending GSCA to pathway analysis of rare variants [18]. Furthermore, Choi et al. (2018) proposed the hierarchical structural component analysis of gene–gene interactions (HisCoM-GGI), an extension of the PHARAOH method, for gene–gene interaction analysis [19]. The HisCoM-GGI method can evaluate both gene-level and SNP-level interactions. The HisCoM-G×E method is an extension of the HisCoM-GGI method for performing gene-based G×E analysis. HisCoM-G×E introduces latent variables, such as a gene, and an environmental factor, which are defined as a weighted sum of the observed variables, such as SNPs, and an environment factor. Accordingly, our proposed method can efficiently account for the biological relationship between a gene and an environmental factor (e.g., alcohol intake), within the structured component.

Previous methods such as GGG tests, used for detecting G×E, integrate multiple *p*-values from each SNP-level interaction test into a single *p*-value of gene-level interactions. By contrast, the HisCoM-G×E method can evaluate G×E effects on the phenotype of interest, all at once. Moreover, our proposed method can account for linkage disequilibrium (LD) between SNPs within a gene, by imposing a ridge penalty. 

In this report, using simulation studies, we show that the performance of the HisCoM-G×E method is similar to or better than other traditional approaches. In addition, we applied the HisCoM-G×E method to a GWAS dataset, as associated with systolic blood pressure (SBP), and an environmental factor (alcohol intake frequency), via genotyping 2252 participants from the Korean Association REsource (KARE) cohort study, using the Affymetrix Genome-Wide Human SNP Array 5.0 [20].

## 2. Results

### 2.1. Type I Error Rate

Appendix A (Appendix A) shows that the empirical type I error rates of the GESAT, iSKAT, GE_minP, GE_GATES, GE_tTS, GE_tProd, and HisCoM-G×E methods were all well preserved, at nominal significance levels for the three scenarios, when the gene size was 5, 50, and 100 SNPs.

### 2.2. Power Comparison

To evaluate the statistical power of the GESAT, iSKAT, GE_minP, GE_GATES, GE_tTS, GE_tProd, and HisCoM-G×E methods, we varied gene and effect sizes of G×E, and then calculated the empirical power for each parameter setting. The results in Figure 1a, when the gene size was 5 SNPs, showed that the GE_GATES method was the most powerful, while the HisCoM-G×E method yielded comparable power. As shown in Figure 1b, when the gene size was 20 SNPs, the iSKAT method was generally the most powerful, and HisCoM-G×E yielded comparable power. In the case of the small effect size of G×E, HisCoM-G×E was more powerful than any of the other methods.

In summary, we can conclude that the HisCoM-G×E method performs reasonably well, regardless of gene size. In other words, it can yield power that is quite comparable to those of the most powerful methods, in the case of different gene sizes.

### 2.3. Real Data Analysis: KARE Dataset

We next applied the iSKAT, GE_GATES, and HisCoM-G×E methods to investigate gene–alcohol intake interactions affecting systolic blood pressure (SBP), using samples from the KARE GWAS dataset [20]. In all, 16,361 genes were included in the analysis. The *p*-values for G×E, using HisCoM-G×E, were calculated via 5000 permutations. In Appendix A (Appendix A), the quantile–quantile (QQ) plot of GE_GATES showed some evidence for deflation of *p*-values, whereas iSKAT and HisCoM-G×E did not. Appendix A (Appendix A) shows the Manhattan plots for the iSKAT, GE_GATES, and HisCoM-G×E methods, where the horizontal red line denotes the threshold for 0.05 genome-wide significance level by a Bonferroni correction of 3.06 × 10^−6^ (α = 0.05/16,361 genes). Table 1 summarizes the 35 G×Es with *p*-values less than the nominal significance level of 0.001. The two genes, *NUCB2* and *ACE*, identified by iSKAT, associate with SBP [21,22,23], and in fact, the angiotensin-converting enzyme (ACE) is a long-held target of numerous antihypertensives [24]. Notably, the HisCoM-G×E method also successfully identified the gene *UGDH*, well known to relate to peripheral arterial disease (PAD) [25]. Additionally, it has been reported that SBP associates with significantly higher risks for a PAD event [26], and that *FCAMR* and *PIGR*, identified by HisCoM-G×E, both significantly associate with coronary atherosclerosis disease [27]. It also has been reported that elevated SBP can predict coronary atherosclerosis, and provides additional information for predicting coronary calcification [28], and is also a risk factor for stroke, myocardial infarction, kidney dysfunction, and aneurysm [29]. Unfortunately, there were no statistically significant interactions detected after Bonferroni correction or the false discovery rate (FDR), using an adjusted *p*-value (*q*-value < 0.05) [30].

## 3. Discussion

In this study, we proposed a new statistical model, namely the Hierarchical structural CoMponent analysis of gene-based Gene–Environment interaction (HisCoM-G×E), to identify gene–environment interactions. Like other hierarchical component models, HisCoM-G×E has several advantages. First, HisCoM-G×E can easily incorporate the biological relationship between a gene and environmental factor. Second, HisCoM-G×E can effectively summarize the interaction effect of a gene and its environment, using structured components. Third, it greatly reduces the dimension of SNPs, in a gene, using a latent variable, defined as a weighted sum of observed variables. In this study, we developed the software for HisCoM-G×E (http://statgen.snu.ac.kr/software/hiscom-gxe). The basic framework of the HisCoM-G×E software is based on the “Workbench for Integrated Superfast Association study with Related Data” (WISARD) [31]. The HisCoM-G×E software was implemented in C++, and was developed for Linux. When our simulation study was executed on a server having two Intel Xeon E5-2620 processors with 128GB of RAM, the HisCoM-G×E program took 30–40 s and 50–60 s when the total number of SNPs in a gene (*K*) was 5 and 20, respectively. Furthermore, HisCoM-G×E can be extended to a statistical method that can take into account various distribution types (i.e., binomial, Poisson, gamma, and inverse Gaussian distributions) by using the framework of generalized linear models (GLMs) applied in the PHARAOH method [18].

One weakness of the study is that the method does not consider epigenetic events (e.g., DNA and histone modifications). Such phenomena are well known to be affected by the environment and exert gene regulation at an additional level, compared to gene sequence alone (e.g., SNPs, indels, copy number, etc.) [32]. To address this shortcoming, future models will be able to consider both genomics and epigenomics in gene regulation/dysregulation.

## 4. Materials and Methods

### 4.1. HisCoM-G×E Method

Let *y_i_* denote the phenotype of the *i*-th subject (*i* = 1, …, *N*). Let *c_ip_* denote the *p*-th covariate of the *i*-th subject (*p* = 1, …, *P*). Let *K* be the number of SNPs in a gene. Let *x_ik_* denote the *k*-th SNP in a gene, for the *i*-th subject (*k* = 1, …, *K*). Let *w_k_* denote a weight assigned to *x_ik_*. Let *g_i_* be a latent variable representing the main effect of a gene, which is defined as a weighted sum of *K* SNPs, such that $g_{i}=\sum_{k=1}^{K}w_{k}x_{\mathrm{ik}}$. Let *z_i_* denote the environment of the *i*-th subject, and *e_i_* be a latent variable representing the environmental effect, which is defined as a weighted sum of environmental effects, such that *e_i_s* = *w_E_z_i_*. Then, the latent interaction term between the gene and environment term is represented by another latent variable, *r_i_*, which is obtained as the product of interacting latent variables, such that $r_{i}=\left(\sum_{k=1}^{K}w_{k}x_{\mathrm{ik}}\right)\cdot \left(w_{E}z_{i}\right){=g}_{i}\cdot e_{i}$.

Let *β*_1_, *β*_2_, *β_p_*, and *β*_12_ denote the coefficients of the gene, the environment, the *p*-th covariate, and latent interaction effects on *y_i_*, respectively. Then, the relationships between the phenotypes and latent variables are established, such that:(1)$$y_{i}{=\beta }_{1}g_{i}{+\beta }_{2}e_{i}{+\beta }_{12}r_{i}+\overset{P}{\underset{p=1}{\sum }}\beta_{p}c_{\mathrm{ip}}{+\varepsilon }_{i},$$
where *ε_i_* is the error term for subject *i*.

Figure 2 shows a simple HisCoM-G×E model with two covariates (i.e., *P* = 2). Rectangles and circles represent observed and latent variables, respectively. For illustrative purposes, we assume that the gene consists of two SNPs (i.e., *K* = 2). The gene, then, is defined as a latent variable constructed by a weighted sum of its SNPs, which in turn influence a phenotype, signified by single-headed arrows. A G×E is similarly defined as a latent variable, constructed via the products of interacting latent variables.

Let ***W*** be a matrix of *w*_1_, *w*_2_, …, *w_k_*, *w_E_*, and ***B*** be a matrix of *β*_1_, *β*_2_, *β_p_*, and *β*_12_. To estimate the unknown parameters for HisCoM-G×E, we adopted the alternating regulated least-squares (ALS) algorithm [18,33]. To estimate unknown parameters in ***W*** and ***B***, we sought to minimize the following Equation (2), subject to the standardization constraint on each latent variable, $\sum_{i=1}^{N}r_{i}^{2}=N$ [34]. Therefore, the penalized least-squares equation is given as follows:(2)$$\phi =SS(y-\hat{y})+\lambda_{S}SS(W)+SS(B),$$
where *λ_S_* is the ridge parameter for the gene. Before applying the algorithm, we used *k*-fold cross-validation (CV) to select the values of *λ_S_* automatically. In our analysis, the ridge parameter, *λ_S_*, is based on five-fold CV, using 15 different starting points of ridge parameters, ranging from 0 to 0.5.

Testing for interactions is equivalent to testing the null hypothesis for the G×E effect, as shown by the following:(3)$$H_{0}{:\;\beta }_{12}=0.$$

We also adopted a permutation procedure, used in our previously developed PHARAOH method [18], to test the statistical significance of the estimated effects of genetics, environment, and the G×E gene–environment. By permuting the phenotype, the HisCoM-G×E method generates the null distributions of weights and path coefficients. Thus, we can obtain empirical *p*-values of interactions between gene and environmental factors.

### 4.2. Simulation Study

To investigate the performance of HisCoM-G×E, we conducted simulation studies to generate human genomic data, with realistic linkage disequilibrium (LD) patterns. We employed a similar simulation strategy that was suggested by Choi et al. [19] for the gene–gene interactions study. In the gene–environment interaction analysis, various factors, i.e., minor allele frequency, haplotype block, and many other factors, can affect the results of a simulation study. In order to reflect various factors, we performed a simulation study by randomly extracting genome data from a real GWAS dataset. In each simulation, we assumed that the phenotype data arose from the following Equation (4), defined as a function of the sum of the genetic value of SNPs, under an additive model, environmental factor, interaction effect, and a random error, following a standard normal distribution, as follows:(4)$$y_{i}{=\beta }_{0}{+\beta }_{1}\left(\sum_{k=1}^{K}w_{k}x_{\mathrm{ik}}\right){+\beta }_{2}e_{i}{+\beta }_{12}\left(\sum_{k=1}^{K}w_{k}x_{\mathrm{ik}}e_{i}\right){+\varepsilon }_{i},$$
where *ε_i_* is the error term for subject *i*.

### 4.3. Type I Error Rate

To check whether the type I error rate is well controlled, we generated datasets under the null hypothesis of no interaction between a gene and its environment. A dataset of *N* samples was generated from the KARE GWAS dataset samples [20]. We considered the following three scenarios, with varying number of SNPs, in a gene: (1) five SNPs (*K* = 5) in *TNFRSF10C*, (2) fifty SNPs (*K* = 50) in *GPR120*, or (3) one hundred SNPs (*K* = 100) in *PLA2G4C*. In Figure 3, the LD pattern plots of those genes were generated using Haploview v4.2 software (Broad Institute, Cambridge, MA, USA) [35].

For each simulated dataset, we randomly selected 500, 1000, and 2000 samples from the KARE GWAS dataset, and generated 1000 simulated datasets, for each scenario. The empirical type I error was estimated based on the significance levels *α* = 0.05, 0.01, and 0.005.

### 4.4. Power Comparison

For power analysis, we varied the number of causal SNPs and the effect sizes of G×E. The performance of HisCoM-G×E was compared to that of GESAT, iSKAT, and the following four GGG methods: GE_minP, GE_GATES, GE_tTS, and GE_tProd [9,14,15]. Two scenarios were considered, with different numbers of SNPs, in various genes as follows: (1) five SNPs (*K* = 5) in *TNFRSF10C* and (2) twenty SNPs (*K* = 20) in *ZNF616*. For each simulated dataset, 500 samples were randomly selected from the KARE GWAS dataset. The parameters were set as follows: total SNPs in a gene (*K*) = 5 and 20, sample size (*N*) = 500, the proportion of causal SNPs = 20%, an effect size, for each SNP (*w_k_*) = 0.5, and the effect size of G×E was as follows: (1) five SNPs (*K* = 5) with *β*_12_ = 0.5, 0.6, 0.7, 0.8, 0.9, or (2) twenty SNPs (*K* = 20) with *β*_12_ = 0.1, 0.2, 0.3, 0.4, 0.5. Since the phenotype was generated by both the effect of SNP (*w_k_*) and the effect of interaction terms (*β*_12_) together, we tested situations in which the effect size was smaller than 1. When the effect size was larger than 1, no difference in the power between statistical methods was observed. The empirical power was estimated by counting the number of associations below a specified significance level (e.g., *α* = 0.05), based on the total number of tests within each set of 1000 simulated datasets.

### 4.5. Real Data Analysis: KARE Dataset

The participants of the KARE project were recruited from Ansung (rural) and Ansan (urban) cohorts in South Korea [20]. Among 8840 subjects, we excluded the subjects with a history of cancer (104 individuals), anti-hypertensive therapy (1019 individuals), drug treatments regulating blood pressure (1055 individuals), and individuals missing values for either body mass index (BMI) (4 individuals) or systolic blood pressure (SBP) (1 individual). Among the female participants (4658 individuals), the alcohol intake frequency was very low, but was relatively high for males, so only 2252 males were included in this study. This study was approved by the Institutional Review Board of the Korean National Institute of Health (IRB No. E1908/001-004). All subjects provided written informed consent to participate in this study, and were then anonymized. We used a detailed genotyping and quality control process previously described [20], and missing genotypes were imputed by IMPUTE2, using Phase 1 of the 1000 Genomes Project as a reference panel [36]. The genetic variants with low IMPUTE2 INFO scores (< 0.8), high missing call rates (> 5%), low minor allele frequencies (≤ 0.01), or of insignificant *p*-values by the Hardy–Weinberg equilibrium test (≤ 1 × 10^−6^) were excluded from the study. Then, we assigned SNPs to genes if they were located within the gene body, or its 20 kb upstream or downstream sequences. As a result, 228,3615 SNPs were mapped to a total of 1,6361 genes. The gene–alcohol intake interaction was analyzed by the HisCoM-G×E method, assuming an additive genetic effect, adjusted for age, recruitment area, and BMI.

## 5. Conclusions

Through the simulation study, we showed that HisCoM-G×E controlled type I error rates well in various scenarios. In power comparison studies, there was only one of the most powerful methods available for the cases of varying gene size. Only HisCoM-G×E yielded power comparable to that of the most powerful methods, regardless of gene size. In the case of the small interaction effect size, HisCoM-G×E yielded the most powerful statistic. Therefore, we recommend HisCoM-G×E as the best choice for analyzing smaller-scale gene–environment interaction studies.

The real data analysis of the KARE GWAS dataset demonstrated that HisCoM-G×E successfully identified genes previously reported for G×E interactions for SBP. Therefore, we fully expect that HisCoM-G×E will help researchers understand environmental contributions to human complex diseases.

## Figures and Tables

**Figure 1 ijms-21-06724-f001:**
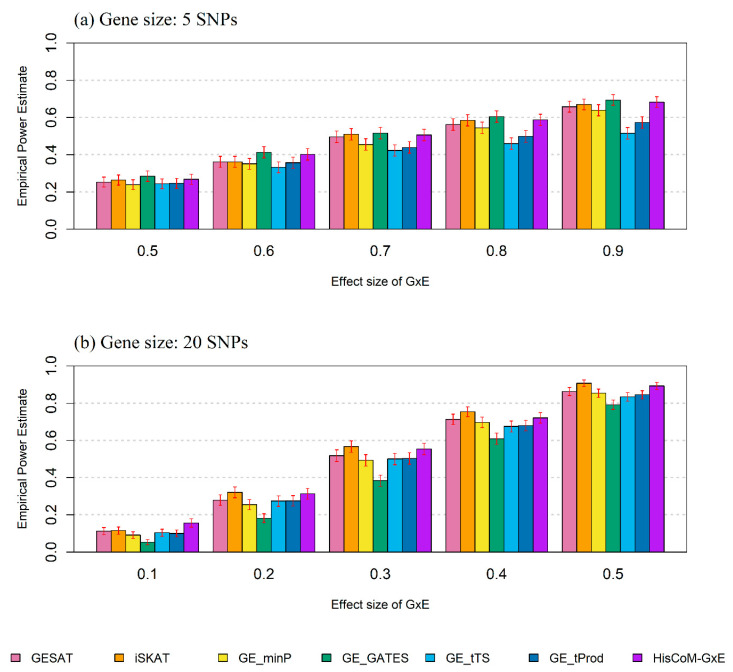
Empirical power estimates as effect size of G×E when the gene sizes are (**a**) 5 SNPs and (**b**) 20 SNPs. The empirical power estimates were calculated with 1000 replicates. Results for each method can be distinguished by plotting colors. Each bar is the mean and the error bars represent standard deviation (SD).

**Figure 2 ijms-21-06724-f002:**
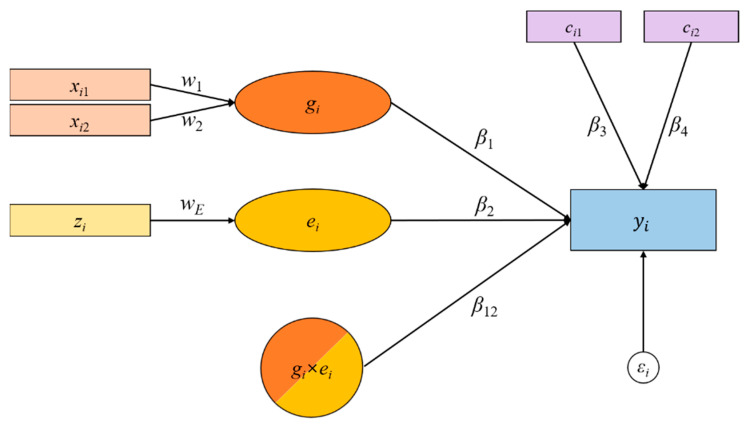
A schematic diagram of Hierarchical structural CoMponent analysis of gene-based Gene–Environment interactions (HisCoM-G×E). The exemplary model is described with the number of single nucleotide polymorphisms (SNPs) (*x_ik_*) *K* = 2 and the number of covariates (*c_ip_*) *P* = 2. The variables *ws* denote the weights assigned to the latent variables and *βs* are coefficients of the latent variables (*g_i_* and *e_i_*). The *g_i_* × *e_i_* term represents a latent interaction term (or effect) on the phenotype (*y_i_*) and *ε_i_* is the error term.

**Figure 3 ijms-21-06724-f003:**
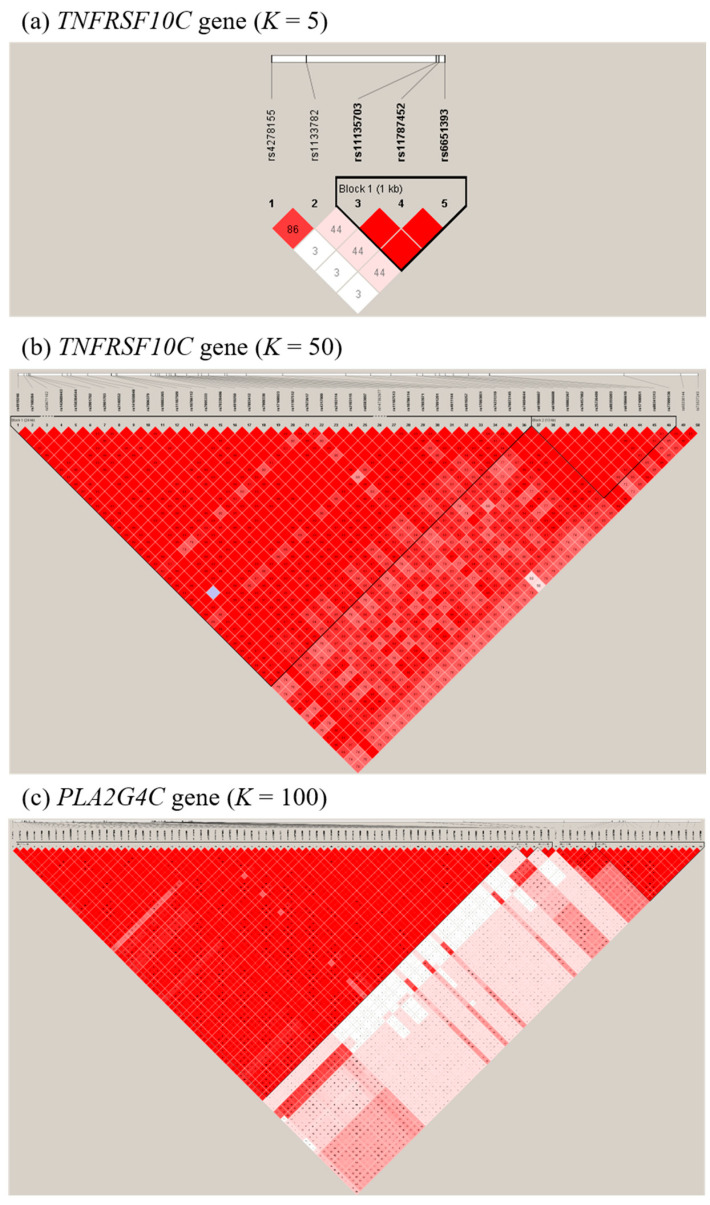
Linkage disequilibrium (LD) patterns of genes used in simulation studies: (**a**) *TNFRSG10C* gene (*K* = 5); (**b**) *TNFRSF10C* gene (*K* = 50); (**c**) *PLA2G4C* gene (*K* = 100). Numbers indicate the *D**ʹ* values expressed as percentiles. A standard color scheme is used to display the LD pattern, with red for perfect LD (*r*^2^ = 1), white for no LD (*r*^2^ = 0) and shades of pink/red for intermediate LD (0 < *r*^2^ < 1).

**Table 1 ijms-21-06724-t001:** Significant (*p* < 0.001; bold) gene–environment interactions that affect systolic blood pressure (SBP), according to the interaction sequence kernel association test (iSKAT), the gene–environment gene-based association test using extended Simes (GE_GATES), and HisCoM-G×E, using the Korea Associated REsource (KARE) genome-wide association studies (GWAS) dataset.

No	CHR	GENE	# of SNPs	iSKAT	GE_GATES	HisCoM-G×E
1	1	*NBPF14*	54	0.5697	0.9902	**6.00 × 10^−5^**
2	19	*TLE*	7	0.2587	0.2491	**1.00 × 10^−4^**
3	4	*COMMD8*	116	0.0285	0.2064	**1.40 × 10^−4^**
4	4	*UGDH*	90	0.0141	0.0518	**2.00 × 10^−4^**
5	8	*ZFAND1*	91	0.1227	0.1713	**2.20 × 10^−4^**
6	2	*RY1*	8	0.1900	0.9966	**2.60 × 10^−4^**
7	9	*GAPVD1*	120	0.4700	0.6905	**2.80 × 10^−4^**
8	1	*FCAMR*	107	0.7512	0.9398	**3.40 × 10^−4^**
9	9	*ADFP*	78	0.6676	0.4259	**4.20 × 10^−4^**
10	1	*PIGR*	119	0.7613	0.9911	**5.00 × 10^−4^**
11	3	*EIF1B*	89	0.1935	0.2530	**8.00 × 10^−4^**
12	19	*ZNF321*	41	**6.74 × 10^−4^**	0.0262	0.0182
13	17	*ACE*	154	**9.50 × 10^−4^**	0.0051	0.0220
14	19	*ZNF566*	11	0.3003	**5.46 × 10^−4^**	0.0670
15	13	*SMAD9*	313	**7.25 × 10^−4^**	0.5510	0.0950
16	8	*LETM2*	107	0.0027	**5.51 × 10^−4^**	0.0974
17	16	*KATNB1*	38	**1.57 × 10^−4^**	0.0011	0.1555
18	16	*PARD6A*	60	**7.36 × 10^−5^**	0.0010	0.1887
19	16	*ACD*	69	**1.08 × 10^−4^**	0.0014	0.2463
20	11	*CALCB*	62	0.8119	**2.45 × 10^−4^**	0.2515
21	14	*RAD51L1*	1199	**2.71 × 10^−4^**	0.1166	0.2515
22	11	*TRIM5*	53	**9.48 × 10^−4^**	0.0070	0.2633
23	16	*C16orf86*	68	**3.30 × 10^−5^**	0.0015	0.2787
24	4	*SORCS2*	1270	**5.99 × 10^−4^**	**2.38 × 10^−5^**	0.3954
25	11	*NUCB2*	134	**8.16 × 10^−5^**	0.0341	0.4232
26	21	*DSCR3*	230	**3.35 × 10^−4^**	0.0060	0.4502
27	16	*RLTPR*	82	**3.30 × 10^−5^**	**8.65 × 10^−4^**	0.4794
28	16	*KIFC3*	131	0.0393	**9.48 × 10^−4^**	0.5110
29	1	*ST6GALNAC3*	1212	**6.51 × 10^−4^**	0.2535	0.5270
30	7	*YKT6*	61	**9.96 × 10^−4^**	0.1369	0.5700
31	19	*ZNF99*	90	**8.90 × 10^−4^**	0.0154	0.5736
32	14	*SSTR1*	40	**7.67 × 10^−4^**	0.0249	0.8029
33	2	*C2orf21*	50	**5.25 × 10^−4^**	0.1123	0.8119
34	8	*WHSC1L1*	134	**5.79 × 10^−4^**	0.0020	0.8379
35	4	*TNFRSF1A*	57	0.5246	**2.57 × 10^−4^**	0.8798

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
