# Peer review of "HisCoM-G×E: Hierarchical Structural Component Analysis of Gene-Based Gene–Environment Interactions"

_ijms, 2020, doi:10.3390/ijms21186724_

Round 1

Reviewer 1 Report

Choi et al. propose a new method for statistical analysis of gene-environment wide association studies, Hierarchical structural CoMponent analysis of gene-based Gene-Environment interactions (HisCoM-G×E). The paper is mostly clear and concise, but I have some questions.

Major reviews

1- Explain the reasoning for selecting TNFRSF10C, GPR120 and PA2G4C to generate datasets under the null hypothesis of no interaction between a gene and its environment and ZNF616 for power comparison. Please provide references if necessary.

2- Line 155, power comparison section - "an effect size, for each SNP (wk) = 0.5". Is not this effect size too big? In quantitative genetics analysis, we should realistically expect for smaller effect sizes. Peharps you intented to set gi = 0.5? But nevertheless it still big. Moreover, the chosen effect size < 1. Have you tested situations in it was > 1? How it could change your results?

3- Still power comparison section - If each SNP has (wk) = 0.5, how could you set 20% of them as causal, since by the previous definition, each SNP has equal effect size? Should not the 20% ones have higher effect size?

4- Still in regards to power - In light of the power comparison results, for which sample size range your HisCoM-G×E method would work best? How would you advise someone intending to use your method for how to calculate said sample size before data collection?

5- Still in regards to power - Discussion section lines 256 - 258: "Only HisCoM-G×E yielded power comparable to that of the most powerful methods, regardless of gene size. In the case of the small interaction effect size, HisCoM-G×E yielded the most powerful statistic.". Therefore your method would be the best choice for analysing smaller-scale gene-environment interaction studies? Please add the responses to this item and the previous item to the Discussion if relevant.

6- Can your methodology be adapted for binary phenotypes (disease/no disease for example)? Please add to Discussion.

7- Discussion, lines 247-248 "As a result, it allows for more efficient computation". Have you performed execution time comparison with the other methods? I suggest that adding a description of your system setup (OS, CPU, RAM, softwares used/required, input data format ...) would also be helpful so other people can use your method in their setting. If not in the main text, add as supplementary data.

Minor reviews

Figure 1. It would be informative if figure legend included a brief description of the meaning of each symbol in the schematics.

Line 133 - "To investigate the performance of HisCoM-G×E, we conducted simulation studies to generate human genomic data, with realistic linkage disequilibrium (LD) patterns." In lieu of reproducibility, could you point some description or references for the method(s) used to perform the simulations?

Lines 146 - 147 "The empirical type-I error was estimated based on the significance levels α = 0.05, 0.01, and 0.05" 0.05 is repeated, correct to 0.005.

Figure 2 - Perhaps the quality/resolution of the figure should be improved.

Figure 5 - The wording of the last sentence of the legend is a bit weird, it could benefit from a rewrite.

In the implementation website, Usage section, the description/instruction for item 1.4 (Option) G×G list File seems to be missing.

Author Response

Major reviews

1- Explain the reasoning for selecting TNFRSF10C, GPR120 and PA2G4C to generate datasets under the null hypothesis of no interaction between a gene and its environment and ZNF616 for power comparison. Please provide references if necessary.

(Response) We appreciate the reviewer’s careful comment. Following the comment, we added the following sentence in ‘2.2. Simulation study’ paragraph.

“We employed a similar simulation strategy that was suggested by Choi et al. [19] for the gene-gene interactions study. In the gene-environment interaction analysis, various factors, i.e. minor allele frequency, haplotype block, and many other factors, can affect the results of simulation study. In order to reflect various factors, we performed a simulation study by randomly extracting genome data from real GWAS dataset.”

2- Line 155, power comparison section - "an effect size, for each SNP (wk) = 0.5". Is not this effect size too big? In quantitative genetics analysis, we should realistically expect for smaller effect sizes. Perhaps you intended to set gi = 0.5? But nevertheless it still big. Moreover, the chosen effect size < 1. Have you tested situations in it was > 1? How it could change your results?

(Response) We appreciate the reviewer’s careful comment. We can agree that the effect size of SNP to 0.5 is assumed to be larger than reality. However, as shown in Equation 4, the phenotype is generating by multiplying the effect of SNP(wk) and the effect of interaction terms (β12) together. Thus, the effect size 0.5 of wk does not represent the direct effect of SNP. Furthermore, we also assumed the effect of SNPs by considering the minor allele frequencies (MAF) of SNPs. In addition, we also tested situations in which the effect size was larger than 1. In this case, the SNP effect size was too large so that there is no difference in power between statistical methods was observed. We added a short paragraph regarding this comment in 2.4. power comparison.

“Since the phenotype was generated by both the effect of SNP (wk) and the effect of interaction terms (β12) together, we tested situations in which the effect size was smaller than 1. When the effect size was larger than 1, there is no difference in the power between statistical methods was observed.”

3- Still power comparison section - If each SNP has (wk) = 0.5, how could you set 20% of them as causal, since by the previous definition, each SNP has equal effect size? Should not the 20% ones have higher effect size?

(Response) We appreciate this comment and are sorry for not providing a clear description on what we compared. When we generated phenotypes, we randomly selected 20% of each SNP (K = 5 or K = 20) and assigned the same value of 0.5, while we assigned the value 0 to the rest SNPs. Of course, the effects of causal SNPs may be all different in reality, but we only focused on the causal SNPs for power comparison.

4- Still in regards to power - In light of the power comparison results, for which sample size range your HisCoM-G×E method would work best? How would you advise someone intending to use your method for how to calculate said sample size before data collection?

(Response) We appreciate the reviewer’s insightful comment. Since the power depends on the number of gene size, minor allele frequency, and LD patterns, it is difficult to make a general statement for the appropriate sample size. However, our limited simulation studies showed that HisCoM-G×E attained enough power when the sample size is as large as 500 for the given simulation setting.

5- Still in regards to power - Discussion section lines 256 - 258: "Only HisCoM-G×E yielded power comparable to that of the most powerful methods, regardless of gene size. In the case of the small interaction effect size, HisCoM-G×E yielded the most powerful statistic.". Therefore, your method would be the best choice for analyzing smaller-scale gene-environment interaction studies? Please add the responses to this item and the previous item to the Discussion if relevant.

(Response) We appreciate the reviewer’s careful comment. Following the reviewer’s comment, we added the following sentence in Discussion.

“Only HisCoM-G×E yielded power comparable to that of the most powerful methods, regardless of gene size. In the case of the small interaction effect size, HisCoM-G×E yielded the most powerful statistic. Therefore, we recommend HisCoM-G×E as the best choice for analyzing smaller-scale gene-environment interaction studies.”

6- Can your methodology be adapted for binary phenotypes (disease/no disease for example)? Please add to Discussion.

(Response) We appreciate the reviewer’s careful comment. Following to the comment, we added the following sentence to Discussion.

“Furthermore, HisCoM-G×E can be extended to a statistical method that can take into account various distribution (i.e. binomial, Poisson, gamma, and inverse Gaussian distributions) by using the frame work of generalized linear models (GLM) applied in the PHARAOH method [18].”

7- Discussion, lines 247-248 "As a result, it allows for more efficient computation". Have you performed execution time comparison with the other methods? I suggest that adding a description of your system setup (OS, CPU, RAM, software used/required, input data format ...) would also be helpful so other people can use your method in their setting. If not in the main text, add as supplementary data.

(Response) We appreciate the reviewer’s careful comment. Following the reviewer’s comment, we rephrased the corresponding sentences in Discussion in order to provide a more detailed description on computation.

“In this study, we developed the software for HisCoM-G×E (http://statgen.snu.ac.kr/software/hiscom-gxe). The basic framework of the HisCoM-G×E software is based on the ‘Workbench for Integrated Superfast Association study with Related Data’ (WISARD) [35]. The HisCoM-G×E software was implemented in C++, and was developed for Linux. When our simulation study was executed on a server having two Intel Xeon E5-2620 processors with 128GB of RAM, HisCoM-G×E program took 30-40 seconds and 50-60 seconds when the total number of SNPs in a gene (K) is 5 and 20, respectively.”

Minor reviews

Figure 1. It would be informative if figure legend included a brief description of the meaning of each symbol in the schematics.

(Response) We appreciate the reviewer’s careful comment. Following the reviewer’s comment, we added the following sentences to Fig.1 legend.

“Figure 1. A schematic diagram of HisCoM-G×E. The exemplary model is described with the number of SNPs (xik) K = 2, the number of covariates (cip) P = 2. The variables ws denote the weights assigned to the latent variables, and βs are coefficients of the latent variables (gi and ei). The gi×ei term represents a latent interaction term (or effect) on the phenotype (yi) and εi is the error term.”

Line 133 - "To investigate the performance of HisCoM-G×E, we conducted simulation studies to generate human genomic data, with realistic linkage disequilibrium (LD) patterns." In lieu of reproducibility, could you point some description or references for the method(s) used to perform the simulations?

(Response) We appreciate the reviewer’s careful comment. Following the reviewer’s comment, we added the following sentence in ‘2.2. Simulation study’ paragraph.

“We employed a similar simulation strategy that was suggested by Choi et al. [19] for the gene-gene interactions study. In the gene-environment interaction analysis, various factors, i.e. minor allele frequency, haplotype block, and many other factors, can affect the results of simulation study. In order to reflect various factors, we performed a simulation study by randomly extracting genome data from real GWAS dataset.”

Lines 146 - 147 "The empirical type-I error was estimated based on the significance levels α = 0.05, 0.01, and 0.05" 0.05 is repeated, correct to 0.005.

(Response) We are sorry for this typo. We modified this sentence.

Figure 2 - Perhaps the quality/resolution of the figure should be improved.

(Response) We agree with the reviewer’s comment. We improved the quality/resolution of Figure 2.

Figure 5 - The wording of the last sentence of the legend is a bit weird, it could benefit from a rewrite.

(Response) We appreciate the reviewer’s careful comment. Following the reviewer’s comment, we modified the following sentence in Fig legend. Furthermore, in response to the request of the other reviewer, we moved the figure to the supplementary figure.

.

“The red horizontal dashed line represents the threshold value of 3.06×10-6 for the 5% genome-wide significance level by Bonferroni correction.”

In the implementation website, Usage section, the description/instruction for item 1.4 (Option) G×G list File seems to be missing.

(Response) We appreciate the reviewer’s careful comment. We deleted item “1.4 (Option) G×G list File” in the HisCoM-G×E website, because it was written incorrectly.

Reviewer 2 Report

The manuscript deals with a fundamental issue in human genomics that in short refers to as Gene-environment interaction (G×E). Often this issue is discussed in view of the “missing heritability” of standard GWAS. The authors present a new GxE detecting statistical method based on Hierarchical structure within a gene (HisCoM-G×E) that they have developed in the past for GxG. Like many of the GWAS based methods the authors try to overcome the inherent difficulty of statistical power associated with SNPs with low effect size.  

The authors nicely cover (in Introduction) the approaches for GxG methods and the associated statistics to justify the expansion to G×E studies. The presented HisCoM-G×E method evaluates the effects on the phenotype while accounting for LD within at a gene resolution. Tests on blood pressure / alcohol intake are presented. The performance of the method by simulations based on the Korean dataset.  A data for downloading is available.

Major comments:

  1. Tests were done by DB with, initially 8,840 subjects ended up with a much a smaller set. The reduction needs to be presented in a clear way (based on lines 170-175). The exclusion and the number associated with each are a bit vague.
  2. Are women excluded from the analysis? Are young people included? A table (in the supplement) with the nature of the DB, the original SNPs within genes, between genes are missing.
  3. Are the differences in Fig 3 significant? Statistical tests should be added to assess the visual differences (if any).
  4. What is the basis for the statement identified genes previously reported for G×E interactions for SBP. Can it be generalized to other traits as presented in Lee et al? Clarification is needed.
  5. The statement for “more efficient computation” should be supported in view of the other methods mentioned.
  6. It is unclear which phenotype is tested Blood pressure? Alcohol intake? Both? Abstract and results are to be consistent.
  7. While it is often the norm to show QQ and Manhattan plot, They are not too informative. They show that all methods failed to find the signal, but each report on numerous noisy loci.

Minor comments:

  1. The notations of Fig. 1 and legend is missing. Much of the ‘text’ needed to be presented as legend. The same for Fig 2 that the LD needs to be explained as well as the color coded and its scale
  2. What is the average number of SNPs associated with each gene (is it typically 20? 50? )
  3. Table 1 is not informative in its current detailed form. Either move to supplement or replace it with compressed graphical representation.
  4. Careless notation on gene names. Is the gene PA2G4C in the text is probably PLA2G4C. The one with n-50 SNPs is it TNFRNF10C (as shown in Fig 2) or is it GPR120 that is mentioned in the text?
  5. Table 2, is it an exhaustive list? What make these genes fully ‘supported’. Are those all genes implicated in SBP?

Author Response

Major comments:

  1. Tests were done by DB with, initially 8,840 subjects ended up with a much a smaller set. The reduction needs to be presented in a clear way (based on lines 170-175). The exclusion and the number associated with each are a bit vague.

(Response) We appreciate the reviewer’s careful comment. Following the reviewer’s comment, we provided a more detailed description on sample exclusion criteria in the revision as follows.

“Among 8,840 subjects, we excluded the subjects with a history of cancer (104 individuals), anti-hypertensive therapy (1,019 individuals), drug treatments regulating blood pressure (1,055 individuals), and individuals missing values for either body mass index (BMI) (4 individuals) or systolic blood pressure (SBP) (1 individual). Among the female participants (4,658 individuals), the alcohol intake frequency was very low, but was relatively high for males, so only 2,252 males were included in this study.”

2. Are women excluded from the analysis? Are young people included? A table (in the supplement) with the nature of the DB, the original SNPs within genes, between genes are missing.

(Response) Following the reviewer’s comment, we provided a more detailed description on the samples. For most 4,568 female participants, the alcohol intake frequencies were not observed, so we excluded women in this study. The KARE cohort recruited the participants over 40 years old, so we did not include young people. When analyzing KARE cohort, in general, data preprocessing and imputation work refer to the same process introduced in Cho et al. (Nat Genet,  2009). Therefore, we provided the detailed descriptions on nature of the DB, the original SNPs, and so on.

3. Are the differences in Fig 3 significant? Statistical tests should be added to assess the visual differences (if any).

(Response) We appreciate the reviewer’s careful comment. In order to compare the difference in the Empirical power estimates for each method, error bars are added in these bar-plots, which makes it possible to distinguish between the most powerful method and the rest of the methods.

4. What is the basis for the statement identified genes previously reported for G×E interactions for SBP? Can it be generalized to other traits as presented in Lee et al? Clarification is needed.

(Response) We appreciate the reviewer’s careful comment. In general, G×E interactions for interesting phenotype are verified by a replication study. However, when the validation dataset is not available, the discovered results (G×E interactions) are also verified by using previously reported results.

5. The statement for “more efficient computation” should be supported in view of the other methods mentioned.

(Response) We appreciate the reviewer’s careful comment. Following the comment, we added the following sentence in Discussion.

“In this study, we developed the software for HisCoM-G×E (http://statgen.snu.ac.kr/software/hiscom-gxe). The basic framework of the HisCoM-G×E software is based on the ‘Workbench for Integrated Superfast Association study with Related Data’ (WISARD) [35]. The HisCoM-G×E software was implemented in C++, and was developed for Linux. When our simulation study was executed on a server having two Intel Xeon E5-2620 processors with 128GB of RAM, HisCoM-G×E program took 30-40 seconds and 50-60 seconds when the total number of SNPs in a gene (K) is 5 and 20, respectively.”

6. It is unclear which phenotype is tested Blood pressure? Alcohol intake? Both? Abstract and results are to be consistent.

(Response) We appreciate the reviewer’s careful comment. Following the comment, we modified the following sentences in Abstract.

“The performance of the proposed method is evaluated in simulation studies, and we applied the proposed method to investigate gene-alcohol intake interactions affecting systolic blood pressure (SBP), using samples from the Korea Associated REsource (KARE) consortium data.”

7. While it is often the norm to show QQ and Manhattan plot, they are not too informative. They show that all methods failed to find the signal, but each report on numerous noisy loci.

(Response) We agree with the reviewer’s point. Since our QQ and Manhattan plots are not so informative, we moved them to supplementary figures.

Minor comments:

  1. The notations of Fig. 1 and legend is missing. Much of the ‘text’ needed to be presented as legend. The same for Fig 2 that the LD needs to be explained as well as the color coded and its scale

(Response) We appreciate the reviewer’s careful comment. Following the comment, we added following sentences in Fig.1 and Fig. 2 legends.

“Figure 1. A schematic diagram of HisCoM-G×E. The exemplary model is described with the number of SNPs (xik) K = 2, the number of covariates (cip) P = 2. The variables ws denote the weights assigned to the latent variables, and βs are coefficients of the latent variables (gi and ei). The gi×ei term represents a latent interaction term (or effect) on the phenotype (yi) and εi is the error term.”

“Figure 2. Linkage disequilibrium (LD) patterns of genes used in simulation studies: (a) TNFRSG10C gene (K = 5); (b) TNFRSF10C gene (K = 50); (c) PLA2G4C gene (K = 100). Numbers indicate the Dʹ values expressed as percentiles. A standard color scheme is used to display the LD pattern, with red for perfect LD (r2 = 1), white for no LD (r2 = 0) and shades of pink/red for intermediate LD (0 < r2 < 1).”

2. What is the average number of SNPs associated with each gene (is it typically 20? 50?)

(Response) The actual number of SNPs in a gene varies much. In real data analysis, the number of SNPs available depends on several factors. For example, depending on the type of GWAS chip (i.e. Affy 5.0, Affy 6.0, or Illumina Omni series), the number of SNPs differs much. In analysis, it also depends on how to define the gene length (i.e. gene body ± 20 kb or ± 50 kb).

3. Table 1 is not informative in its current detailed form. Either move to supplement or replace it with compressed graphical representation.

(Response) We appreciate the reviewer’s careful comment. Following the comment, we moved the table to the supplementary file.

4. Careless notation on gene names. Is the gene PA2G4C in the text is probably PLA2G4C. The one with n-50 SNPs is it TNFRSNF10C (as shown in Fig 2) or is it GPR120 that is mentioned in the text?

(Response) We are sorry for this error. There are 5 SNPs in TNFRSF10C gene and 50 SNPs in GPR120 gene. We modified the legend of Figure 2 as follows.

“Figure 2. Linkage disequilibrium (LD) patterns of genes used in simulation studies: (a) TNFRSG10C gene (K = 5); (b) TNFRSF10C gene (K = 50); (c) PLA2G4C gene (K = 100). Numbers indicate the Dʹ values expressed as percentiles. A standard color scheme is used to display the LD pattern, with red for perfect LD (r2 = 1), white for no LD (r2 = 0) and shades of pink/red for intermediate LD (0 < r2 < 1).”

5.  Table 2, is it an exhaustive list? What make these genes fully ‘supported’. Are those all genes implicated in SBP?

(Response) The 35 G×Es listed in Table 2 are candidate GxEs expected to be associated with SBP through iSKAT, GE_GATES, and HisCoM-G×E. Among them, biological meaning was found trough literature reviews, two genes (NUCB2 and ACE genes) through iSKAT, and three genes (UGDH, FCAMR and PIGR genes) through HisCoM-G×E.

Round 2

Reviewer 2 Report

The authors addressed the major comments in a satisfactory way. The current Figures and tables are appropriate and the information in the supplemental is completed.